# Chain-of-Glimpse: Search-Guided Progressive Object-Grounded Reasoning for Video Understanding

Zhixuan Wu [* 1]   Quanxing Zha [* 2]   Teng Wang [3]   Genbao Xu [4]   Wenyuan Gu [1]
Wei Rao [5]   Nan Ma [4]   Cheng Bo [1]   Soujanya Poria [5]

## Abstract

Video understanding requires identifying and reasoning over semantically discriminative visual objects across frames, yet existing object-agnostic solutions struggle to effectively handle substantial object variations over time. To address this, we introduce **Chain-of-Glimpse**, a search-guided progressive object-grounded reasoning framework that explicitly anchors each reasoning step to specific visual evidence regions, enabling compositional and multi-step decision-making. Formally, Chain-of-Glimpse formulates video reasoning as a step-by-step process that incrementally builds spatially grounded traces around task-relevant visual objects, thereby mitigating over-reliance on saliency-driven cues. Specifically, Chain-of-Glimpse features a search-guided controller, optimized via reinforcement learning with a format reward that significantly incentivizes grounding capability, to iteratively ground visual evidence regions and form reliable reasoning trajectories, yielding accurate and interpretable multi-step decisions. Extensive evaluations across two categories of video reasoning benchmarks, including general video reasoning benchmarks such as NExT-QA, Video-Holmes, CG-Bench-Reasoning, and VRBench, and grounded video reasoning benchmarks such as NExT-GQA, demonstrate that Chain-of-Glimpse consistently improves performance while exhibiting strong robustness and generalization across diverse video reasoning tasks.

[*]Equal contribution [1]State Key Laboratory of Networking and Switching Technology, Beijing University of Posts and Telecommunications, Beijing, China. [2]Institute of Big Data, College of Computer Science and Artificial Intelligence, Fudan University, China. [3]ARC Lab, Tencent PCG, Shenzhen, China. [4]School of Artificial Intelligence, Beijing University of Technology, Beijing, China. [5]Nanyang Technological University, Singapore. Correspondence to: Cheng Bo <chengbo@bupt.edu.cn>, Soujanya Poria <soujanya.poria@ntu.edu.sg>.

*Proceedings of the $43^{rd}$ International Conference on Machine Learning*, Seoul, South Korea. PMLR 306, 2026. Copyright 2026 by the author(s).

## 1. Introduction

Video understanding (Zhang et al., 2024a; Guo et al., 2025; Tao et al., 2025) has emerged as a cornerstone of multi-modal learning, requiring models to jointly reason over visual and linguistic context to support complex decision-making. Despite recent successes, some advanced models (Huang et al., 2024; Ventura et al., 2025) still rely predominantly on text-based reasoning, with their cognitive processes remaining largely confined to the language modality. In contrast, human reasoning is inherently object-grounded, progressively integrating visual evidence with contextual and linguistic cues to form coherent interpretations. This cognitive gap motivates a shift toward models that more closely emulate human-like multi-modal reasoning (Zhang et al., 2024b; Zhao et al., 2025b; Sarch et al., 2025). Milestone models such as the Qwen-VL (Wang et al., 2024a) have successfully integrated vision-and-language processing by aligning fine-grained visual features with linguistic tokens, thereby enhancing the reasoning process through a seamless fusion of modalities. By enabling the localization of specific image regions, these works have brought multi-modal large language models (MLLMs) closer to the object-centric nature of human cognition, partially bridging the gap between passive perception and active reasoning within the static spatial domain.

Extending this fine-grained grounding to the temporal domain, several advancements (Ren et al., 2024; Han et al., 2025; Bai et al., 2025; Feng et al., 2025b) have begun to extend MLLMs for sequence-level reasoning. To bridge the gap between model capabilities and human-like cognition, existing approaches can be broadly categorized into two paradigms. First, vanilla RL-based methods (Feng et al., 2025a; Li et al., 2025b) have shown that reinforcement learning (RL) can effectively enhance reasoning capabilities by enabling models to iteratively refine decisions based on feedback, as shown in Fig. 1(a). However, these approaches typically operate under a single-step perception regime, where the visual encoding is performed once and the subsequent reasoning lacks dynamic, multi-step interaction with visual evidence. Second, CoT-based methods (Zheng et al., 2023; Han et al., 2025; Li et al., 2025a; Zhong et al.,

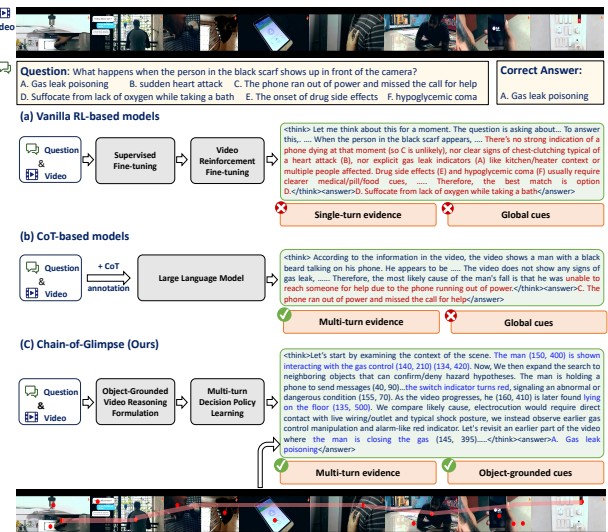

*Figure 1.* **Inconsistent reasoning in prior models and improved with Chain-of-Glimpse.** (a) **Vanilla RL-based** and (b) **CoT-based models** both *insufficient evidence integration* and *global context oversight*, as they tend to rely on *superficial, visually prominent cues*. Consequently, they fail to capture complex dependencies, leading to inconsistent reasoning (D and C). In contrast, (c) our **Chain-of-Glimpse** performs progressive, object-grounded reasoning through a *spatio-temporal dynamic chain* across objects. Our method effectively synthesizes multi-turn evidence with global awareness for accurate and robust predictions.

2025) have adopted step-by-step reasoning to mimic logical progression, as shown in Fig. 1(b). Nevertheless, they generally rely on static inference frameworks and remain fundamentally language-centric, lacking explicit grounding in fine-grained visual evidence during the reasoning process. As shown in Fig. 1, both paradigms struggle to anchor their predictions on subtle yet critical visual evidence. In this case, despite the presence of a task-critical object indicating a gas leak, prior models fail to ground their reasoning on this evidence and instead produce divergent conclusions. This may lead to biased reasoning, as it may overlook objects that are more semantically relevant to the question, resulting in inaccurate conclusions.

Motivated by these insights, we propose **Chain-of-Glimpse**, a search-guided progressive object-grounded reasoning framework for video understanding that endows MLLMs with the ability to "think with videos" by actively seeking visual evidence. To capture the complex dynamics of video content, we design a object-grounded video reasoning paradigm that constructs spatio-temporal trajectories. This paradigm enables the model to actively navigate through the video, modeling both intra-frame object semantics and inter-frame temporal correlations. By explicitly selecting and attending to task-relevant objects across frames, the model builds a coherent evidence chain that grounds abstract reasoning in concrete visual details. To ensure the reliability

of these trajectories, we introduce a multi-turn decision policy learning. Rather than relying on naive reinforcement learning optimization, this mechanism aligns step-wise exploration with final task objectives, stabilizing the reasoning process. It encourages the model to suppress visually salient but semantically irrelevant cues, ensuring that each decision step contributes meaningfully to the final prediction. As a result, Chain-of-Glimpse bridges low-level visual perception and high-level logical reasoning, producing decision paths that are both accurate and interpretable.

Collectively, the main contributions of this paper can be summarized as:

- We construct an object-grounded video reasoning formulation (OGVRF) that models video reasoning as a multi-step process over object-level representations. This establishes a principled foundation for progressive reasoning, allowing to explicitly ground task-relevant visual evidence across frames while simultaneously mitigating over-reliance on saliency-driven cues.

- We introduce a multi-turn decision policy (MTDP) that formulates multi-step reasoning as a stochastic trajectory optimization problem. Through policy optimization with designed grounding-aware reward, MTDP encourages MLLMs to iteratively explore and consolidate high-reward reasoning trajectories, thereby supporting interpretable and self-correcting decisions grounded in object-level evidence over long horizons.

- We conduct extensive experiments across general and grounded video reasoning benchmarks, demonstrating consistent and significant performance gains as well as generalization across diverse video reasoning tasks.

## 2. Related Work

### 2.1. Multi-modal Large Language Models

Multi-modal large language models (MLLMs) have recently advanced multi-modal video reasoning and shown strong performance on tasks, such as video question answering (videoQA) (Jin et al., 2024; Min et al., 2024) and video captioning (videoCap) (Seo et al., 2022; Shu et al., 2025). Existing MLLMs (Liu et al., 2023; Qian et al., 2024) extract frame- or clip-level visual representations using pre-trained vision encoders and align them with textual embeddings to enable cross-modal interaction. However, most approaches rely on coarse temporal aggregation or uniform frame sampling, which limits their ability to capture fine-grained temporal dynamics and structured visual information. To improve their efficiency, several studies (Song et al., 2025; Liu et al., 2025b) introduce hierarchical or memory-based designs that process videos at multiple temporal scales. While these methods improve temporal dependency modeling, they

often introduce substantial computational overhead and remain vulnerable to noise caused by redundant or irrelevant visual content. Moreover, many MLLMs (Tang et al., 2023; Hu et al., 2025) depend on large amounts of task-specific supervision or curated instruction datasets, which constrains their generalization to unseen scenarios and real-world applications. Despite these advances, existing MLLMs (He et al., 2024; Mitra et al., 2024) still face challenges in effectively modeling structured visual information and leveraging it for reliable multi-modal reasoning. We address these limitations by explicitly modeling object-level semantics and interactions across video frames, enabling progressive reasoning over temporally aligned visual cues.

## 2.2. Object-grounded Reasoning

Object-grounded reasoning (Yuan et al., 2025b; Cai et al., 2025; Ye et al., 2025) has emerged as an effective paradigm for enhancing multi-modal understanding by explicitly modeling object-level semantics and their interactions. Rather than treating visual inputs as holistic frame representations, object-grounded approaches (Fei et al., 2024; Yuan et al., 2025a) decompose scenes into structured entities and relations, enabling more fine-grained perception and reasoning over complex visual content. This paradigm is particularly beneficial for video reasoning, where object dynamics and interactions evolve over time. Several studies (Huang et al., 2025; Sarch et al., 2025) incorporate reinforcement learning (RL) to guide object-centric perception and reasoning processes. Such designs have been applied to visual question answering and video reasoning, where an agent iteratively refines its reasoning policy based on task feedback. Although RL-based approaches (Feng et al., 2025a; Zheng et al., 2025) improve interpretability and adaptability, they often require carefully designed reward functions and substantial training signals, which limits their scalability and stability in large-scale multi-modal settings. More recently, chain-of-thought (CoT) reasoning (Tang et al., 2023; Zhao et al., 2025a) has emerged as an effective paradigm for enhancing the reasoning capability of large language models by decomposing complex tasks into intermediate reasoning steps. In multi-modal contexts, CoT-based approaches (Chen et al., 2024b) extend this idea by aligning visual observations with step-by-step textual reasoning, enabling models to perform compositional inference over object attributes and relationships. Object-aware CoT reasoning (Fei et al., 2024; Shao et al., 2024) has been explored for multi-step logical inference, where intermediate reasoning states help bridge low-level visual perception and high-level semantic understanding. However, existing methods often rely on implicit object representations or heuristic prompting strategies, making it challenging to ensure consistent grounding between visual objects and textual reasoning chains. Our approach addresses these limitations by combining object-grounded

representations with multi-turn decision policies and reinforcement learning, enabling models to explicitly select and verify object-level evidence across video frames.

## 3. Preliminary

The overarching objective of our work is to study object-grounded reasoning with multi-modal large language models (MLLMs)in videos, aiming to enable models to explicitly select and verify object-level evidence across time. Given a video input $V = \{v_t\}_{t=1}^{T}$, where $T$ frames are sampled from the raw stream, and a corresponding textual instruction $L = \{l_n\}_{n=1}^{N}$ consisting of $N$ tokens, we extract per-frame object instances $O_t = \{o_{t,m}\}_{m=1}^{M_t}$ via generic detection, where $M_t$ denotes the total number of objects in frame $t$. Object-grounded reasoning maintains a query-conditioned reasoning state $h^{(k)}$ at step $k$ that summarizes previously selected evidence and the multi-modal context. Selecting evidence is formulated as a multi-step decision process, represented as a sequence of object-level selections $a_{1:K}$. We define an autoregressive policy over this sequence:

$$\Pi_\theta(a_{1:K}|V, L) = \prod_{k=1}^{K} \Pi_\theta(a_k|a_{<k}, V, L), \qquad (1)$$

where $a_k$ denotes the selection of object-level evidence at step $k$, and $K$ denotes the maximum number of evidence selection steps. While supervised learning can be applied when step-wise selections are annotated, such annotations are often unavailable for long videos or open-ended queries. Therefore, we consider the objective of maximizing the expected return over trajectories induced by $\Pi_\theta$:

$$\max_\theta J(\theta) = \mathbb{E}_{a_{1:K} \sim \Pi_\theta(\cdot|V,L)}[R(V, L, a_{1:K})], \qquad (2)$$

where $R(\cdot)$ is a task-level reward that may reflect grounding quality, encouraging the model to suppress salient-but-irrelevant objects while emphasizing task-relevant yet low-saliency evidence in the reasoning process.

## 4. Method

To tackle the challenge of reliable video reasoning under complex temporal dynamics, we propose **Chain-of-Glimpse**, search-guided progressive object-grounded reasoning for video understanding, as shown in Fig. 2. Rather than treating video reasoning as a one-shot mapping from visual inputs to answers, Chain-of-Glimpse reformulates reasoning as a sequential decision-making process, where the model incrementally selects, verifies, and aggregates object-level evidence over time. Specifically, Chain-of-Glimpse instantiates this paradigm through two tightly coupled components: (i) **Object-grounded video reasoning formulation**, which defines video reasoning as a sequence of

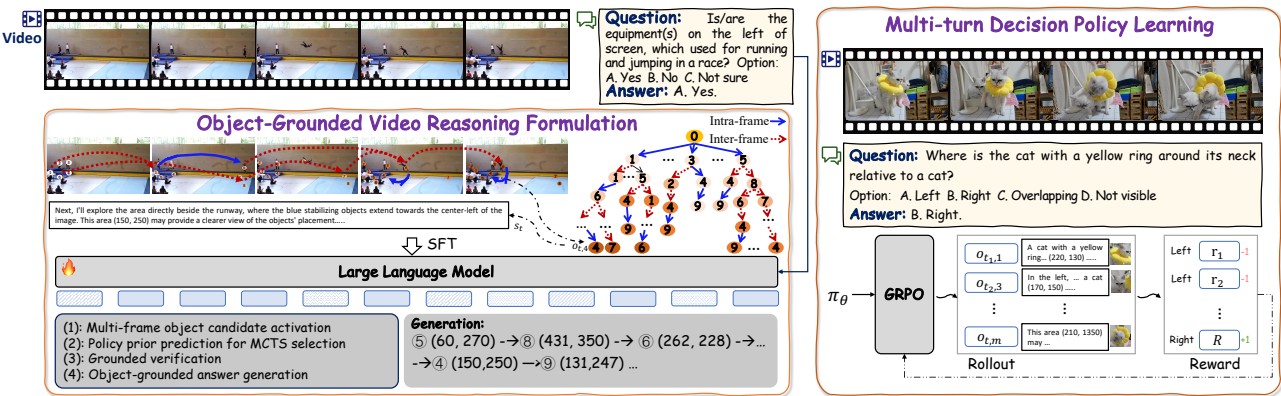

Figure 2. **Overview of Chain-of-Glimpse.** Chain-of-Glimpse formulates video reasoning as a search-guided, multi-turn object-grounded decision process. Given a video and a query, the model searches over object-grounded reasoning trajectories and optimizes them via reinforcement learning with task-level rewards, enabling accurate reasoning beyond visually salient cues.

object-centric reasoning states and leverages explicit search over reasoning trajectories to generate high-quality, verifiable evidence paths for supervised initialization; (ii) **Multi-turn decision policy learning**, which optimizes reasoning policies under sparse, task-level rewards, enabling stable multi-step reasoning, reducing reliance on visually salient but semantically irrelevant cues, and enforcing consistent object-grounded decision-making across turns.

### 4.1. Object-Grounded Video Reasoning Formulation

We introduce an **o**bject-**g**rounded **v**ideo **r**easoning **f**ormulation (OGVRF) that casts video understanding as a progressive, multi-step decision process grounded in object-level visual evidence. Unlike state-of-the-art MLLMs that predominantly rely on single-pass generation or implicitly encoded reasoning traces, OGVRF explicitly models reasoning as a sequence of grounded transitions anchored in object-level evidence spanning multiple objects and multiple frames. More details on OGVRF in Appendix A.1.

**MCTS-based Reasoning Trajectory Search.** In contrast to traditional autoregressive reasoning paradigms that commit to a single reasoning path, we employ Monte Carlo Tree Search (MCTS) (James et al., 2017) to explore multiple object-grounded reasoning trajectories. This design allows the model to systematically balance exploration and exploitation, mitigating early commitment to visually salient yet task-irrelevant evidence. By expanding the search tree iteratively, the model is able to evaluate different reasoning paths and adaptively refine its reasoning process. We find that most candidates can be easily identified as negative samples without requiring complex reasoning. To improve efficiency, we define a reasoning state $h^{(k)} = f(h^{(k-1)}, a_k, v_{t(a_k)})$ that summarizes the accumulated reasoning history up to step $k$, $t(a_k)$ denotes the frame index associated with action $a_k$, $f(\cdot)$ denotes a state transi-

tion function. At each reasoning step, the available actions are defined over detected objects conditioned on the current reasoning state $\mathcal{A}(h^{(k)}) = \bigcup_{t \in \mathcal{T}(h^{(k)})} \mathcal{P}(O_t)$, where $\mathcal{P}(\cdot)$ represents valid object selections, and $\mathcal{T}(h^{(k)})$ specifies the candidate frame indices reachable under the current state. Each action $a_k \in \mathcal{A}(h^{(k)})$ corresponds to selecting one or more object instances from the per-frame object set $O_{t(a_k)}$, leading to a successor state $h^{(k+1)}$. A reasoning trajectory corresponds to a path $\tau = \{h^{(0)}, a_1, h^{(1)}, a_2, \ldots, h^{(K)}\}$ in the search tree, $h^{(0)}$ is initialized from the instruction $L$ and the global video context, $K$ denotes the total number of reasoning steps. During tree traversal, action selection follows a policy-guided upper confidence bound:

$$
\begin{aligned}
a_k^* = \arg\max_{a_k} &\left[ Q\Big(h^{(k)}, a_k\Big) \right. \\
&\left. + \lambda P_\theta\Big(a_k \mid h^{(k)}\Big) \frac{\sqrt{\sum_b N\big(h^{(k)}, b\big)}}{1 + N\big(h^{(k)}, a_k\big)} \right].
\end{aligned}
\tag{3}
$$

where $Q(h^{(k)}, a_k) = W(h^{(k)}, a_k)/N(h^{(k)}, a_k)$ denotes the estimated action value, $W(\cdot, \cdot)$ is the accumulated backed-up value. $P_\theta(a_k|h^{(k)})$ is a learned policy prior, $N(h^{(k)}, a_k)$ is the visit count, $\lambda$ controls the exploration strength, $b \in \mathcal{A}(h^{(k)})$ enumerates all actions available at state $h^{(k)}$. This formulation enables multi-step reasoning across both objects and frames, allowing the model to evaluate multiple candidate trajectories, discard low-value branches early, and revise decisions through backtracking.

**Grounded Object-State Transition.** To adequately integrate video grounding into the search process, each state transition in the reasoning tree is explicitly grounded in temporally indexed object instances. At each reasoning step $k$, an action is defined as the selection of an object instance:

$$
a_k = (t_k, m_k), \quad o_{t_k, m_k} \in O_{t_k},
\tag{4}
$$

where $t_k$ denotes the frame index at step $k$ and $m_k$ indexes the object within that frame $O_{t_k}$. The reasoning history is updated via an object-grounded transition function:

$$h^{(k+1)} = \phi(h^{(k)}, \mathbf{f}_{(t_k, m_k)}) \tag{5}$$

where $\phi(\cdot)$ denotes the state transition function, $\mathbf{f}_{(t_k, m_k)}$ represents the visual features of the selected object at time $t_k$, encoding both appearance and temporal context within the video. To preserve temporal coherence, we enforce a monotonic constraint on frame indices $t_{k+1} \geq t_k$, ensuring that reasoning progresses forward in time while still allowing the search process to revisit previously grounded objects through alternative branches in the search tree.

Once the MCTS search phase has identified the potential reasoning paths, we distill these trajectories into the model via supervised fine-tuning (SFT). Specifically, we collect the top-scoring trajectories into a supervision set $\mathcal{D}_{\mathrm{mcts}} = \{(h^{(k)}, a_k)\}$ where each trajectory contributes multiple state–action pairs. The loss is then formulated as:

$$\mathcal{L}_{\mathrm{SFT}}(\theta) = -\mathbb{E}_{(h^{(k)}, a_k) \sim \mathcal{D}_{\mathrm{mcts}}} \left[ \log \pi_\theta(a_k \mid h^{(k)}) \right], \tag{6}$$

which corresponds to maximizing the likelihood of the MCTS-selected actions under the policy parameterized by $\theta$. Minimizing this loss produces the reference policy $\pi_{\mathrm{ref}} \triangleq \pi_\theta$, which serves as both the initialization and a stable behavioral prior for subsequent policy optimization.

## 4.2. Multi-turn Decision Policy Learning

To fully exploit the potential of object-grounded reasoning in complex video tasks, we introduce **m**ulti-**t**urn **d**ecision **p**olicy (MTDP) learning. Existing approaches often prioritize visually salient frames and cues during inference, which can result in suboptimal decisions when task-relevant but less prominent objects are overlooked. MTDP addresses this limitation by modeling multi-turn reasoning as a sequence of decisions conditioned on task objectives, enabling the model to systematically explore and select relevant object-level evidence across steps, independent of visual prominence.

**Policy Optimization.** Unlike previous models that primarily focus on saliency, which can lead to decisions based on visually dominant but irrelevant objects, group relative policy optimization (GRPO) aims to balance exploration and exploitation in the reasoning process. For each training sample $(V, L)$ we sample a group of $G$ object-grounded reasoning trajectories $\{a_{1:K}^{(j)}\}_{j=1}^G$ from the current policy $\pi_\theta$ where each trajectory follows the same object-level state transitions defined in OGVRF and induces a sequence of reasoning states $\{h^{(j,k)}\}_{k=1}^K$. Each trajectory is evaluated with a scalar task-level reward $R_j \triangleq R(V, L, a_{1:K}^{(j)})$. We compute a group-relative advantage without a critic:

$$A_j = \frac{R_j - \mu_R}{\sigma_R + \epsilon}, \tag{7}$$

where $\mu_R$ and $\sigma_R$ denote the mean and standard deviation of rewards within the group. The policy is updated using a clipped importance-weighted objective with KL regularization to a reference policy $\pi_{ref}$, following standard PPO-style optimization. The loss can be formulated as:

$$\mathcal{L}_{\mathrm{GRPO}}(\theta) = -\frac{1}{G} \sum_{j=1}^G A_j \frac{\pi_\theta(a_{1:K}^{(j)} \mid V, L)}{\pi_{\mathrm{ref}}(a_{1:K}^{(j)} \mid V, L)} \tag{8}$$
$$+ \beta \, \mathrm{KL}[\pi_{\mathrm{ref}} \parallel \pi_\theta],$$

minimizing which updates the policy to favor high-reward trajectories while remaining close to the reference policy.

**Reward Design.** The reward function is designed to align evidence selection with task-level correctness rather than visual prominence. We use a composite reward:

$$R(V, L, a_{1:K}^{(j)}) = R_{\mathrm{ans}}(V, L, a_{1:K}^{(j)})$$
$$+ \alpha \frac{1}{K} \sum_{k=1}^K R_{\mathrm{evid}}\left(h^{(j,k)}, a_k^{(j)}\right), \tag{9}$$

where $\alpha$ controls the trade-off between answer correctness $R_{\mathrm{ans}}$ and quality of object-level evidence $R_{\mathrm{evid}}$.

## 4.3. Learning and Inference Procedure

After introducing the proposed new methods, we then present the overall process for model training and inference in this section. The OGVRF is first used to extract object-level representations and define object-grounded transitions. Candidate reasoning trajectories are then explored using MCTS and distilled into the model via supervised fine-tuning in Sec. 4.1. Finally, we optimize the policy using MTDP in Sec. 4.2, where challenging samples on which the SFT model fails to produce accurate results with trajectory-level rewards that jointly capture answer correctness and evidence-grounding quality. During inference, the model generates a small set of object-grounded trajectories and selects the best trajectory to produce the final answer.

# 5. Experiments

## 5.1. Datasets and Protocols

We construct our training and evaluation data based on the SA2VA dataset (Yuan et al., 2025a). To comprehensively assess Chain-of-Glimpse, we evaluate it across both general and grounded video reasoning benchmarks. Specifically, we use NExTQA (Xiao et al., 2021) for general video reasoning and NExT-GQA (Xiao et al., 2024) for grounded video reasoning. To further assess generalization, we adopt additional external benchmarks, including Video-Holmes (Cheng et al., 2025), CG-Bench-Reasoning (Chen et al., 2025a), and VR-Bench (Yu et al., 2025), which differ substantially from SA2VA in content and reasoning requirements. We follow

*Table 1.* **Comparison on NExTQA benchmarks.** *Note:* MC indicates multiple-choice, '–' indicates the result is not reported.

| Method | Backbone | Avg | MC | | |
|---|---|---|---|---|---|
| | | | Causal | Temporal | Descriptive |
| LLoVi (Zhang et al., 2024a) | GPT-3.5 | 66.3 | 67.1 | 60.1 | 76.5 |
| VideoAgent (Wang et al., 2024b) | GPT-4 | 71.3 | 72.7 | 64.5 | 81.1 |
| DyCoke (Tao et al., 2025) | LLaVA-OV-0.5B | 57.2 | – | – | – |
| DyCoke (Tao et al., 2025) | LLaVA-OV-7B | 79.4 | – | – | – |
| LangRepo (Kahatapitiya et al., 2025) | Mixtral 8×7B | 60.9 | – | – | – |
| Video-LLaVA (Lin et al., 2024) | Vicuna-v1.5-7B | 60.5 | 61.2 | 54.2 | 71.1 |
| Qwen2.5-VL-3B (Bai et al., 2025) | Qwen2.5-VL-3B | 72.9 | 73.3 | 67.6 | 81.9 |
| **Chain-of-Glimpse-3B** | Qwen2.5-VL-3B | 73.6 | 73.9 | 68.8 | 81.6 |
| Qwen2.5-VL-7B (Bai et al., 2025) | Qwen2.5-VL-7B | 79.7 | 79.8 | 77.6 | 83.3 |
| **Chain-of-Glimpse-7B** | Qwen2.5-VL-7B | 83.3 | 83.7 | 80.4 | 87.1 |

*Table 2.* **Comparison across proprietary and open-source models on Video-Holmes, CG-Bench-Reasoning and VRBench.** SR stands for Social Reasoning; IMC stands for Intention & Motive Chaining; TCI stands for Temporal Causal Inference; TA stands Timeline Analysis; MHR stands for Multi-modal Hint Reasoning; PAR stands for Physical Anomaly Reasoning; CTI stands for Core Theme Inference. CG-Bench-Reasoning reports Long-Acc, VRBench reports multiple-choice (MCQ) accuracy.

| Method | Size | Frames | Video-Holmes | | | | | | | | CG-Bench | VRBench |
|---|---|---|---|---|---|---|---|---|---|---|---|---|
| | | | Avg | SR | IMC | TCI | TA | MHR | PAR | CTI | | |
| *Proprietary Models* | | | | | | | | | | | | |
| GPT-4o (Hurst et al., 2024) | – | 32 | 42.0 | 50.0 | 49.6 | 38.8 | 30.0 | 44.0 | 39.2 | 37.0 | 45.2 | 81.8 |
| OpenAI o4-mini (Hurst et al., 2024) | – | 32 | 29.9 | 36.3 | 31.2 | 20.5 | 34.0 | 30.1 | 30.9 | 27.4 | 33.4 | 73.9 |
| Gemini-2.0-Flash (Comanici et al., 2025) | – | – | 30.6 | 41.8 | 33.7 | 23.1 | 20.5 | 30.1 | 26.8 | 33.7 | - | - |
| *Open-source Models* | | | | | | | | | | | | |
| InternVL2.5-8B (Chen et al., 2024a) | 8B | 32 | 23.8 | 28.0 | 32.2 | 21.5 | 7.7 | 25.7 | 23.8 | 22.6 | 19.4 | 26.7 |
| InternVL3-8B (Chen et al., 2024a) | 8B | 32 | 32.3 | 29.5 | 40.7 | 37.9 | 35.1 | 24.6 | 38.9 | 24.1 | 24.2 | - |
| Qwen2.5-Omni-7B (Bai et al., 2025) | 7B | 32 | 16.4 | 27.1 | 19.9 | 13.9 | 7.5 | 14.8 | 14.9 | 13.7 | 23.9 | - |
| VideoChat-R1 (Li et al., 2025b) | – | 32 | 33.0 | 42.1 | 38.8 | 24.5 | 39.5 | 29.5 | 27.8 | 29.3 | - | - |
| SEED-Bench-R1 (Chen et al., 2025b) | – | 32 | 33.5 | 42.8 | 35.1 | 25.6 | 40.5 | 29.2 | 29.9 | 32.6 | - | - |
| Qwen2.5-VL-3B (Bai et al., 2025) | 3B | 32 | 34.5 | 48.6 | 33.6 | 35.9 | 28.6 | 35.3 | 32.9 | 28.2 | 32.1 | 25.2 |
| **Chain-of-Glimpse-3B** | 3B | 32 | 36.0 | 48.3 | 40.2 | 30.0 | 30.5 | 34.0 | 30.0 | 35.2 | 33.6 | 26.3 |
| Qwen2.5-VL-7B (Bai et al., 2025) | 7B | 32 | 38.8 | 53.4 | 43.5 | 30.4 | 32.5 | 36.1 | 35.1 | 37.4 | 34.3 | 46.0 |
| **Chain-of-Glimpse-7B** | 7B | 32 | 42.5 | 56.8 | 43.8 | 32.4 | 37.5 | 37.8 | 36.8 | 38.5 | 36.2 | 47.5 |

the standard evaluation protocol of each benchmark. Additional implementation details are provided in Appendix B.2.

## 5.2. Implementation Details

All experiments are conducted on 3 NVIDIA A800 GPUs. We adopt Qwen2.5-VL-7B, initialized from the pretrained weights released in LLaMAFactory (Zheng et al., 2024), as the base MLLM. OGVRF is trained for 30 epochs with a batch size of 8 and a learning rate of 1e-6. MTDP is trained for 500 steps with a learning rate of 1e-6, using 8 rollouts per sample. To bootstrap high-quality object-grounded reasoning trajectories, GPT-4o is employed only during the MCTS-based data generation stage to expand reasoning nodes and produce grounded reasoning steps with object coordinates. The resulting trajectories are then used for supervised fine-tuning and policy optimization of Qwen2.5-VL-7B (Bai et al., 2025). Importantly, GPT-4o is not used during training or inference, ensuring that all reported re-

sults are produced solely by the target MLLM. During training, the vision encoder remains frozen in all stages, and the language model is fine-tuned using LoRA to ensure training efficiency. We use Qwen2.5-VL-7B as the baseline network and conduct ablation experiments to study the effects across NExTQA, NExT-GQA and Video-Holmes three benchmarks (see more details in Appendix B.2).

## 5.3. Main Results

We evaluate the effectiveness of Chain-of-Glimpse on the NExTQA benchmarks, as shown in Tab. 1, including proprietary LLM-based pipelines and open-source MLLMs. To assess the scalability and generalization of our approach, we report results for both Chain-of-Glimpse-3B and Chain-of-Glimpse-7B. Under the same backbone setting, our Chain-of-Glimpse-7B consistently improves over the baseline Qwen2.5-VL-7B, achieving 83.3% *v.s.* 79.7%. In terms of category-wise multiple-choice performance,

*Table 3.* Ablation results for different reasoning strategies. *Long* denotes videos with $\geq$1,000 frames, whereas *Short* denotes videos with <1,000 frames. naive RL uses Vanilla GRPO.

| Variant | MCTS | SFT | GRPO | NexTQA | | | Video-Holmes |
| --- | --- | --- | --- | --- | --- | --- | --- |
| | | | | Avg. | Long | Short | |
| Baseline | ✗ | ✗ | ✗ | 79.7 | 77.9 | 81.2 | 38.8 |
| + reasoning (CoT) | ✗ | ✗ | ✗ | 80.4 | 79.1 | 81.7 | 38.6 |
| + Object-Grounded (Random) | ✗ | ✗ | ✗ | 79.7 | 77.1 | 82.4 | 38.8 |
| + Object-Grounded (Greedy) | ✗ | ✗ | ✗ | 80.9 | 79.4 | 82.3 | 39.1 |
| + SFT | ✗ | ✓ | ✗ | 80.6 | 78.6 | 82.5 | 39.3 |
| + SFT + GRPO | ✗ | ✓ | ✓ | 81.4 | 80.1 | 82.9 | 40.6 |
| + OGVRF | ✓ | ✓ | ✗ | 82.3 | 82.2 | 82.5 | 40.9 |
| + OGVRF + naive RL | ✓ | ✓ | ✗ | 82.7 | 83.5 | 81.9 | 41.8 |
| Full (ours) | ✓ | ✓ | ✓ | 83.3 | 84.2 | 82.4 | 42.5 |

Chain-of-Glimpse-7B yields gains on both Causal (83.7% *v.s.* 79.8%) and Temporal (80.4% *v.s.* 77.6%), indicating that the improvement mainly comes from better handling of causal/temporal dependencies rather than shortcutting easy descriptive cues. Notably, despite relying on an open-source 7B-scale backbone, Chain-of-Glimpse-7B surpasses GPT-based systems such as VideoAgent (71.3%) and LLoVi (66.3%), highlighting the effectiveness of our reasoning and verification strategy.

Tab. 2 compares proprietary and open-source models on three general video reasoning benchmarks: Video-Holmes, CG-Bench-Reasoning, and VRBench. GPT-4o achieves strong overall performance with 42.0% average accuracy, but exhibits notable imbalance across reasoning categories, particularly in temporal causal inference (TCI 38.8%, TA 30.0%), indicating limited explicit temporal decision modeling. Among open-source baselines, Qwen2.5-VL-7B attains an average of 38.8%, but shows weaker performance on intention and motive chaining (IMC 43.5%) and temporal causal inference (TCI 30.4%). By contrast, Chain-of-Glimpse consistently improves over its backbone at both 3B and 7B scales. At 3B, it increases the average score from 34.5% to 36.0% and improves IMC (+6.6) and CTI (+7.0), while also achieving higher CG-Bench Long-Acc (33.6% *v.s.* 32.1%). At 7B, the gains are more pronounced, raising the average to 42.5% (+3.7) and improving ITA (37.5% *v.s.* 32.5%), TCI (32.4% *v.s.* 30.4%), and MHR (37.8% *v.s.* 36.1%). These consistent gains across benchmarks and reasoning categories indicate that progressive object-grounded decision modeling substantially improves out-of-domain generalization. More discussion is provided in Appendix C.

### 5.4. Ablation Studies

**Effect of different components.** To verify the effectiveness of each component, Tab. 3 reports results on NExTQA and Video-Holmes. Adding SFT further stabilizes predictions

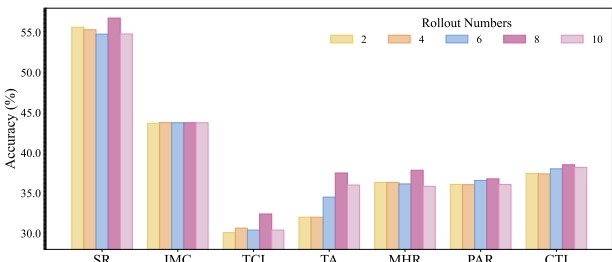

*Figure 3.* Effect of MCTS rollout numbers on Video-Holmes.

by providing reference trajectories distilled from search-guided reasoning, boosting NExTQA to 80.6% and Video-Holmes to 39.3%. GRPO-based reinforcement fine-tuning optimizes multi-turn reasoning policies, yielding additional gains (NExTQA 81.4%, Video-Holmes 40.6%). Incorporating OGVRF enhances long-video performance (NExTQA 82.3%), and the full model achieves the best results (NExTQA 83.3%, Video-Holmes 42.5%), demonstrating the complementary benefits of each component.

**Effect of different object exploration strategies.** Tab. 3 compares random and greedy object exploration without MCTS, SFT, or GRPO. Random grounding provides no gain over the baseline, Greedy exploration yields consistent improvements, increasing NExTQA average to 80.9% and Long accuracy to 79.4%, and improving Video-Holmes to 39.1%. It also outperforms CoT on NExTQA and Video-Holmes, highlighting the importance of selecting informative objects and motivating structured search in OGVRF.

**Effect of MCTS rollout numbers.** To study the effect of search depth, we vary the number of MCTS rollouts during reasoning trajectory search. Fig. 3 shows the impact of different MCTS rollout numbers on Video-Holmes. Our results indicate that a moderate number of rollouts consistently improves performance on reasoning categories that require complex temporal and object-level reasoning, such as tem-

*Table 4.* Ablation on MCTS rollouts and inference cost on NExT-GQA.

| Method | Rollouts | Acc@GQA | Acc@QA | Inf. Time |
|---|---|---|---|---|
| Qwen2.5-VL-7B | 0 | – | 74.4 | 2.29 |
| Chain-of-Glimpse-7B | 2 | 30.5 | 75.1 | 2.30 |
| Chain-of-Glimpse-7B | 4 | 30.9 | 76.5 | 2.35 |
| Chain-of-Glimpse-7B | 6 | 31.4 | 77.2 | 2.37 |
| Chain-of-Glimpse-7B | **8** | **32.1** | **77.9** | 2.37 |
| Chain-of-Glimpse-7B | 10 | 30.9 | 77.5 | 3.01 |

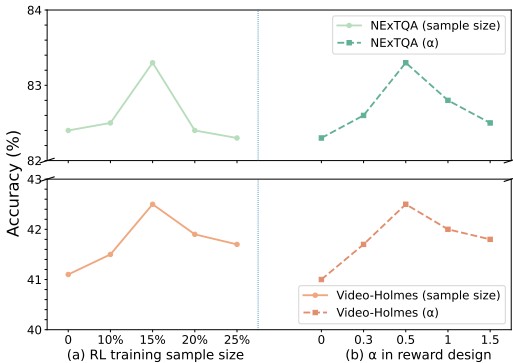

*Figure 4.* Ablation studies on NExTQA and Video-Holmes.

poral causal inference (TCI), trajectory alignment (TA), and multi-hop relational reasoning (MHR). This improvement arises because additional rollouts allow the model to explore alternative object-grounded trajectories and better integrate evidence across frames. Overall, these findings highlight that balancing exploration depth is critical for enhancing multi-step object-grounded reasoning without introducing oversearching or unnecessary computational overhead.

**Effect of inference cost.** We further analyze inference efficiency on NExT-GQA. As shown in Tab. 4, Qwen2.5-VL-7B requires 2.29 s/sample, while our methods with 8 rollouts achieves the best performance (32.1 Acc@GQA, 77.9 Acc@QA) with only 2.37 s/sample, corresponding to a marginal overhead of 0.08s. This demonstrates that our method introduces only limited additional cost compared to a strong baseline. Moreover, we observe that increasing the number of rollouts from 2 to 8 consistently improves performance ($30.5 \rightarrow 32.1$ Acc@GQA, $75.1 \rightarrow 77.9$ Acc@QA) while maintaining nearly constant inference time ($2.30 \rightarrow 2.37$ s/sample). Further increasing rollouts to 10 leads to higher latency (3.01 s/sample) without performance gains, indicating that a small rollout budget is sufficient in practice. In addition, the reasoning trajectory has a fixed and shallow depth, which keeps the search space bounded and prevents the computational cost from growing with the number of candidate objects or video length.

**Effect of RL training sample size.** Fig. 4(a) studies how the amount of RL data affects performance. Compared with using no RL data, incorporating a small portion of samples already yields consistent gains, indicating that GRPO effectively refines the policy beyond SFT initialization. Performance improves as the RL data increases and peaks around 15%-20%, while further scaling to 25% brings marginal benefit or slight degradation, suggesting diminishing returns and potential overfitting. We adapt 15% as the default setting for a favorable accuracy-efficiency trade-off.

**Effect of $\alpha$ in reward design.** Fig. 4(b) studies the impact of $\alpha$, which balances the final answer reward $R_{ans}$ and the step-level evidence reward $R_{evid}$. We observe that introducing a moderate evidence reward consistently improves performance over $\alpha = 0$, indicating that step-wise grounding helps the policy learn more reliable evidence chains.

However, overly large $\alpha$ leads to slight degradation, suggesting that excessive emphasis on intermediate evidence may distract the model from optimizing final answer correctness. $\alpha = 0.5$ yields the best trade-off and is used by default.

**Effect of Policy Optimization** Tab. 3 analyzes the impact of different policy optimization on video understanding. Introducing RL without explicit grounding (SFT + GRPO) brings limited gains over the baseline, improving NextTQA from 79.7% to 81.4% and Video-Holmes from 38.8% to 40.6%, indicating that answer-level optimization alone is insufficient for complex video reasoning. We use Vanilla GRPO as naive RL, which when applied to OGVRF does not produce stable gains, indicating that standard RL objectives are not well aligned with multi-turn grounded reasoning. Chain-of-Glimpse combines OGVRF with a carefully designed reward optimizing both answer correctness and evidence quality achieves the best results with 83.3% on NexTQA and 42.5% on Video-Holmes, demonstrating that RL is effective only when tightly coupled with object-grounded reward design.

**Effect of video length.** Tab. 3 shows that video length substantially affects performance on NExTQA. The baseline exhibits a clear gap between long and short videos, with 77.9% on Long and 81.2% on Short. In contrast, OGVRF substantially strengthens long-video understanding, increasing Long accuracy to 83.2% while maintaining comparable performance on Short at 81.6%. Chain-of-Glimpse-7B further improves Long accuracy to 84.2% and Short accuracy to 82.4%, achieving the best overall average accuracy of 83.3%. These results indicate that progressive object-grounded decision policies are particularly effective for long-horizon reasoning, where models must accumulate, validate, and integrate evidence over extended temporal contexts.

### 5.5. Visualization

We present a case study to aid an intuitive understanding of the superiority of our system. As shown in Fig. 5, our

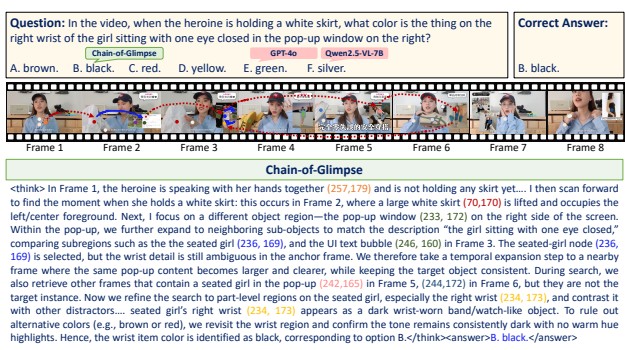

*Figure 5.* Visualization of qualitative example showcasing how our Chain-of-Glimpse achieves successful video understanding.

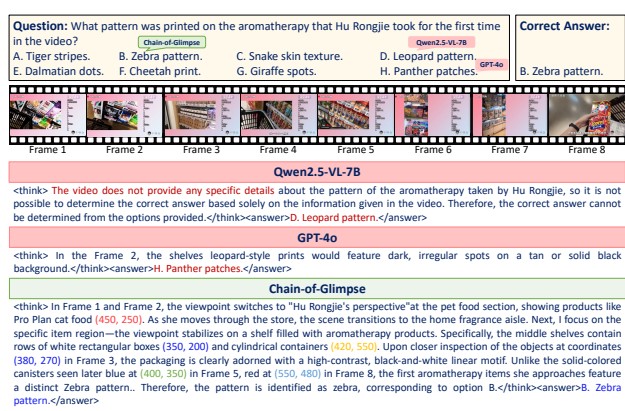

*Figure 6.* Qualitative visualizations.

Chain-of-Glimpse correctly answers black, whereas GPT-4o and Qwen2.5-VL-7B predict green and silver, respectively. A closer inspection reveals that the baselines fail to maintain consistent object grounding across reasoning steps. In particular, GPT-4o appears to attend to an incorrect frame, mistakenly associating the queried wrist attribute with another girl appearing earlier in the video, while Qwen2.5-VL-7B selects an irrelevant accessory due to ambiguous visual focus. In contrast, our Chain-of-Glimpse explicitly decomposes the problem into a sequence of grounded decisions, progressively narrowing down the correct temporal segment, visual context, and target object before attribute recognition. This structured reasoning process not only leads to the correct answer but also ensures that each intermediate step is visually and semantically interpretable. In the Appendix D, we show more qualitative visualizations of examples.

As shown in Fig. 6, our proposed method Chain-of-Glimpse demonstrates strong performance on fine-grained visual discrimination and object-centric reasoning in complex video scenarios. In the cases presented, the qualitative results highlight the limitations of standard VLMs. While Qwen2.5-VL-7B fails to extract sufficient visual details to produce a conclusive answer, and GPT-4o misinterprets the visual pattern by incorrectly associating it with leopard-style fea-

tures, our Chain-of-Glimpse method correctly identifies the Zebra pattern (Option B). This improvement stems from the Chain-of-Glimpse mechanism, which explicitly performs sequential, goal-directed visual evidence selection over time, enabling the model to verify object-level cues across frames rather than relying on single-pass holistic representations.

## 6. Conclusions

In this work, we introduce Chain-of-Glimpse, a search-guided progressive object-grounded reasoning framework for video understanding. Chain-of-Glimpse enables stable multi-turn decision-making over temporally grounded object evidence, effectively mitigating inconsistent predictions induced by single-step global representations and saliency-driven shortcuts. Extensive experiments on NExT-QA, Video-Holmes, NExT-GQA, CG-Bench-Reasoning, and VRBench validate the effectiveness and superiority of Chain-of-Glimpse across diverse video reasoning benchmarks with varying content distributions and reasoning requirements. We benchmark Chain-of-Glimpse against both proprietary models, such as GPT-4o, and open-source models, such as Qwen2.5-VL-7B, highlighting the superiority of our approach. Furthermore, detailed analyses quantify the contributions of different design strategies and confirm the capacity of Chain-of-Glimpse to perform stable, multi-turn, object-grounded reasoning over extended temporal contexts.

## Acknowledgments

We would like to thank the anonymous reviewers, our program chairs, senior area chairs, area chairs for their thoughtful comments and support on this work. This work was supported by the National Natural Science Foundation of China (Grant No.62372058, U22A2026).

## Impact Statement

This paper presents work whose goal is to advance the field of Machine Learning, with a particular focus on large language models (LLMs) and multi-modal large language models (MLLMs) for video understanding. By explicitly modeling reasoning over object instances across time, the proposed approach seeks to improve the transparency, reliability, and reasoning capabilities of multi-modal systems in video question answering and related tasks.

The potential societal impact of this work includes enhancing the robustness and reliability of AI systems in applications such as assistive technologies, video analysis, and human–computer interaction. There are many potential societal consequences of our work, none which we feel must be specifically highlighted here.

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

# A. Extended Details of Methodology

## A.1. Phases of MCTS-Based Reasoning Trajectory Search

We summarize the Monte Carlo Tree Search (MCTS) procedure used in OGVRF for object-grounded video reasoning. The search systematically explores reasoning trajectories across both frames and object instances. Starting from an initial reasoning state $h^{(0)}$, the search iteratively selects actions, where each action $a_k = (t_k, m_k)$ corresponds to choosing an object instance from frame $t_k$. Selection at each step is guided by a learned policy prior and value estimates, balancing exploration of new trajectories and exploitation of high-value paths. Upon expansion, the reasoning state is updated with the visual features of the selected object, encoding both appearance and temporal context. The trajectory value is then backpropagated along the visited path to update estimates for ancestor nodes, allowing the search to revise earlier decisions. The process continues until a fixed budget or maximum depth is reached, resulting in a set of high-scoring object-grounded trajectories for supervised fine-tuning or inference. In this appendix, we provide a detailed step-by-step description of the MCTS procedure used in OGVRF.

**Selection.** Starting from the root node, the search tree is traversed iteratively by selecting actions $a_k \in \mathcal{A}(h^{(k)})$ according to a policy-guided upper confidence bound in Eq. 3.

**Expansion.** Once a leaf node is reached, a new child node is created by selecting an object-grounded action $a_k = (t_k, m_k)$ and updating the reasoning state in Eq. 5.

**Simulation.** We directly utilize the policy prior $P_\theta(a_k \mid h^{(k)})$ to estimate the value of the newly expanded node, reflecting the expected quality of the reasoning trajectory.

**Backpropagation.** The estimated trajectory value is propagated backward along the visited path to update the $Q$ values and visit counts of ancestor nodes, allowing the search to revise earlier decisions based on new evidence.

**Termination.** The search continues until a fixed budget of expansions or a maximum depth $K$ is reached. The top-scoring trajectories are collected to form a supervision set for SFT.

## A.2. Formulations of Group Relative Policy Optimization

We provide a detailed formulation of the Group Relative Policy Optimization (GRPO) method used for optimizing object-grounded reasoning policies. Given a set of $J$ candidate reasoning trajectories $\{\tau_j\}_{j=1}^J$ generated via MCTS or other proposal mechanisms, each trajectory $\tau_j = \{h^{(0)}, a_1, h^{(1)}, \ldots, a_K, h^{(K)}\}$ is associated with a cumulative reward $R(\tau_j)$ that jointly evaluates answer correctness and object-level evidence grounding quality. GRPO treats the top-performing trajectory as a reference and encourages the policy to assign higher probability mass to actions along trajectories that achieve superior rewards relative to other candidates. Formally, let $\pi_\theta(a_k \mid h^{(k)})$ denote the policy for selecting object-grounded actions. The relative policy optimization objective is defined as:

$$\mathcal{L}_{\text{GRPO}} = -\frac{1}{J} \sum_{j=1}^{J} \sum_{k=1}^{K} \left( \Delta R_j \, \log \pi_\theta(a_k^j \mid h^{(k),j}) \right), \tag{10}$$

where $\Delta R_j = R(\tau_j) - \bar{R}_{\text{group}}$, $\bar{R}_{\text{group}} = \frac{1}{J} \sum_{i=1}^{J} R(\tau_i)$ measures the relative quality of trajectory $\tau_j$ against the average reward $\bar{R}_{\text{group}}$ of the candidate group. Positive $\Delta R_j$ encourages the policy to reinforce actions along high-reward trajectories, while negative $\Delta R_j$ suppresses suboptimal decisions. This formulation naturally generalizes standard policy gradient methods by considering relative rewards within a group of trajectories rather than absolute returns, improving the efficiency of trajectory-level credit assignment and stabilizing learning in multi-step, object-grounded video reasoning. During optimization, trajectories can be filtered or weighted based on length, frame coverage, or grounding quality to further emphasize high-quality reasoning paths.

# B. More Details of Experiments

## B.1. Details of Training Data

As discussed in Sec. 5.1 of the main paper, before using learning multi-turn decision policies for further optimization, we leverage GPT-4o to synthesize a object-grounded reasoning dataset from SA2VA to activate the reasoning capabilities of the MLLM. Given a video $V = \{v_t\}_{t=1}^T$, the extracted object instances $O_t = \{o_{t,m}\}_{m=1}^{M_t}$, and a language instruction $L$, we

construct multi-turn decision trajectories that explicitly select object-level evidence across time and progressively refine the reasoning state until producing the final answer.

**Step 1: Object-grounded reasoning trajectory speculation.** We use GPT-4o to generate a speculative description of what an ideal result should look like. The prompt encourages GPT-4o to: (i) identify candidate objects relevant to the query, (ii) iteratively select object instances as evidence, and (iii) update the reasoning state until a final answer is produced. Each speculative trajectory consists of a sequence of object-grounded actions and a final answer, forming an initial reasoning trace that serves as a high-level guide for search-based exploration. The generated trajectory description is denoted as $\hat{\tau}$.

**Step 2: MCTS-based object-level trajectory expansion.** Based on the speculative trajectory $\hat{\tau}$, we perform Monte Carlo Tree Search (MCTS) to explore alternative object-grounded reasoning paths. Each node in the search tree represents a reasoning state conditioned on $(V, L)$ and the previously selected object instances, while each edge corresponds to selecting a new object-grounded action.

**Step 3: Trajectory filtering and normalization.** The raw MCTS-generated trajectories are further filtered to ensure data quality and consistency. Specifically, we remove trajectories that: (i) lead to incorrect or unverifiable final answers, (ii) repeatedly select irrelevant or redundant object instances, or (iii) exceed a predefined maximum reasoning length. For the remaining trajectories, we normalize the output format into a unified multi-turn template, where each step contains the selected object identifier, a concise evidence statement, and the updated reasoning state. This normalization step ensures stable supervision signals across different samples and tasks.

**Step 4: OGVRF data construction.** After filtering and normalization, the retained trajectories are converted into supervised fine-tuning samples. Each SFT example consists of the original input $(V, L)$ followed by the full multi-turn decision trace and the final answer. This dataset teaches the model to imitate structured object-grounded decision policies, encouraging it to explicitly select and justify object-level evidence rather than relying on implicit visual saliency or single-pass reasoning. The resulting dataset is denoted as $\mathcal{D}_{\text{SFT}}$ and is used to initialize the policy model.

**Step 5: MTDP data construction.** Following the setup in Fig. 4(a) of the main paper, the total number of MTDP trajectories is set to approximately 15% of the OGVRF dataset size. This dataset is used exclusively for reinforcement learning to further optimize the policy toward improved answer correctness and object-level evidence fidelity, while leveraging the SFT policy as a behavioral prior.

### B.2. Details of Implementation

Our training framework is implemented using the `LlamaFactory` codebase (Zheng et al., 2024) and executed on a cluster of 3 NVIDIA A800 GPUs. We perform experiments on both the 3B and 7B variants of the `Qwen2.5-VL` model family. Throughout both the OGVRF and MTDP stages, the vision tower remains frozen to preserve stable spatio-temporal representations.

For the OGVRF stage, we apply Low-Rank Adaptation (LoRA) to all linear modules with a rank $r = 128$ and a scaling factor $\alpha = 256$. The model is trained on the SA2VA dataset for 30 epochs using the AdamW optimizer with a peak learning rate of $1.0 \times 10^{-6}$ and a cosine decay schedule with a 0.03 warmup ratio. The sequence length is set to 8,192 tokens. In the MTDP stage, we optimize the policy using the GRPO objective with $J = 8$ rollouts per prompt. This stage employs a constant learning rate of $1.0 \times 10^{-6}$ and a KL-divergence penalty coefficient $\beta = 0.01$ to regularize the policy against the SFT baseline. For experience collection, we limit prompt and response lengths to 1,024 tokens each, using a sampling temperature of 0.5 for training and a decoding temperature of 1.0 for evaluation. The optimization is performed with a maximum gradient norm of 1.0. Training terminates upon reaching the predefined rollout-update iteration limit, with generation governed by specific termination tokens (`</tool_call>`, `</answer>`, `<|im_end|>`).

## C. More Results

### C.1. More Comparison on EgoSchema and STAR bechmarks

To further assess out-of-domain generalization, we report results on EgoSchema (Mangalam et al., 2023) and STAR (Wu et al., 2024) in Table 5, both of which emphasize egocentric perception and multi-step temporal reasoning. On EgoSchema, Chain-of-Glimpse shows consistent improvements over Qwen2.5-VL under identical model sizes and frame budgets. The 3B model improves accuracy from 33.3% to 34.6%, while the 7B model achieves a larger gain from 60.0% to 62.2%. On STAR, Chain-of-Glimpse yields higher average performance across both model scales. For the 7B setting, the average score

*Table 5.* Comparison on EgoSchema test set and the STAR validation set under identical model sizes and frame settings.

| Method | Size | Frames | EgoSchema | STAR | | | | |
|---|---|---|---|---|---|---|---|---|
| | | | | Avg | Feasibility | Interaction | Prediction | Sequence |
| Qwen2.5-VL-3B (Bai et al., 2025) | 3B | 32 | 33.3 | 65.4 | 68.0 | 62.1 | 71.5 | 70.1 |
| **Chain-of-Glimpse-3B** | 3B | 32 | 34.6 | 66.8 | 66.9 | 62.2 | 68.3 | 69.5 |
| Qwen2.5-VL-7B (Bai et al., 2025) | 7B | 32 | 60.0 | 70.0 | 70.0 | 64.6 | 72.3 | 73.2 |
| **Chain-of-Glimpse-7B** | 7B | 32 | 62.2 | 71.1 | 71.8 | 64.9 | 72.8 | 74.9 |

increases from 70.0% to 71.1%, with improvements observed in Feasibility from 70.0% to 71.8%, Interaction from 64.6% to 64.9%, and Sequence from 73.2% to 74.9%. The 3B model also improves the average score from 65.4% to 66.8% while maintaining comparable performance across all sub-tasks.

### C.2. More Comparison on NExT-GQA bechmarks

we have extended our evaluation to the temporally and spatially grounded benchmark NExT-GQA. As shown in Tab. 6, our method achieves 32.1 Acc@GQA, while remaining competitive performance on localization metrics (41.0 mIoP and 31.4 mIoU). These results suggest that the observed improvements stem from more precise object-level and temporally grounded reasoning, rather than superficial answer matching.

*Table 6.* Comparison on NExT-GQA bechmarks.

| Method | Acc@GQA | mIoP | mIoU |
|---|---|---|---|
| LangRepo (8×7B) (Liu et al., 2025a) | 17.1 | 31.3 | 18.5 |
| VideoChat-TPO-7B (Yan et al., 2025) | 25.5 | 35.6 | 27.7 |
| VideoMind-7B (Liu et al., 2025a) | 28.2 | 39.0 | 31.4 |
| **Chain-of-Glimpse-7B** | 32.1 | 41.0 | 31.4 |

### C.3. Generalization Across MLLM Families on NExTQA and NExT-GQA benchmarks

To examine whether Chain-of-Glimpse is tied to a specific backbone architecture, we further extend our experiments to additional MLLM families, including InternVL3-8B and LLaVA-NeXT-Video-7B. As reported in Tab. 7, Chain-of-Glimpse consistently improves performance across all evaluated backbones. With InternVL3-8B as the backbone, it improves the average accuracy on NExTQA from 77.3 to 81.3, and increases Acc@QA on NExT-GQA from 72.8 to 75.6. Similarly, with LLaVA-NeXT-Video-7B, Chain-of-Glimpse improves the NExTQA accuracy from 75.5 to 78.2, and increases Acc@QA on NExT-GQA from 71.3 to 72.8. These results indicate that the proposed Chain-of-Glimpse formulation is not restricted to the Qwen series, but transfers effectively to heterogeneous multimodal architectures. The consistent gains across Qwen, InternVL, and LLaVA-NeXT-Video further suggest that the improvements primarily arise from the proposed object-grounded reasoning mechanism, rather than architecture-specific characteristics of a particular model family.

### C.4. Computational efficiency comparison on NExTQA benchmark

As shown in Table 8, Chain-of-Glimpse-7B with Qwen2.5-VL-7B achieves the highest accuracy of 83.3% while requiring only 6.47 TFLOPs. Compared with DyCoke (14.13 TFLOPs, 79.4%) and FastVID (10.73 TFLOPs, 77.0%), our method achieves better performance with substantially lower computational cost. These results demonstrate that Chain-of-Glimpse provides a favorable accuracy–efficiency trade-off for video reasoning.

## D. More Qualitative Visualizations

We present a failure case in Fig. 7 where Chain-of-Glimpse correctly localizes the cup of water and tracks the artist's hand across frames but predicts the incorrect functional intent. The model concludes that the cup's purpose is to provide moisture for the paint (option A), while the ground-truth answer is to clean the paintbrush (option E). This shows that accurate object grounding alone is insufficient for fine-grained functional reasoning. The model relies on plausible but shallow causal cues

*Table 7.* **Generalization across different MLLM backbones on NExTQA and NExT-GQA.**

| Method | Backbone | NExTQA (Avg.) | NExT-GQA (Acc@QA) |
|---|---|---|---|
| Qwen2.5-VL-7B* | Qwen2.5-VL-7B | 79.7 | 74.4 |
| Chain-of-Glimpse-7B | Qwen2.5-VL-7B | **83.3** | **77.9** |
| InternVL3-8B* | InternVL3-8B | 77.3 | 72.8 |
| Chain-of-Glimpse-8B | InternVL3-8B | **81.3** | **75.6** |
| LLaVA-NeXT-Video-7B* | LLaVA-NeXT-Video-7B | 75.5 | 71.3 |
| Chain-of-Glimpse-7B | LLaVA-NeXT-Video-7B | **78.2** | **72.8** |

\* indicates results reproduced using the official open-source code.

*Table 8.* **Computational efficiency comparison on NExTQA benchmark.**

| Method | Backbone | TFLOPs | Avg. Acc. (%) |
|---|---|---|---|
| DyCoke (Tao et al., 2025) | LLaVA-OV-7B | 14.13 | 79.4 |
| FastVID (Shen et al., 2026) | LLaVA-OV-7B | 10.73 | 77.0 |
| Chain-of-Glimpse-7B | Qwen2.5-VL-7B | 6.47 | 83.3 |
| Chain-of-Glimpse-8B | InternVL3-8B | 7.08 | 82.7 |
| Chain-of-Glimpse-7B | LLaVA-NeXT-Video-7B | 5.93 | 80.3 |

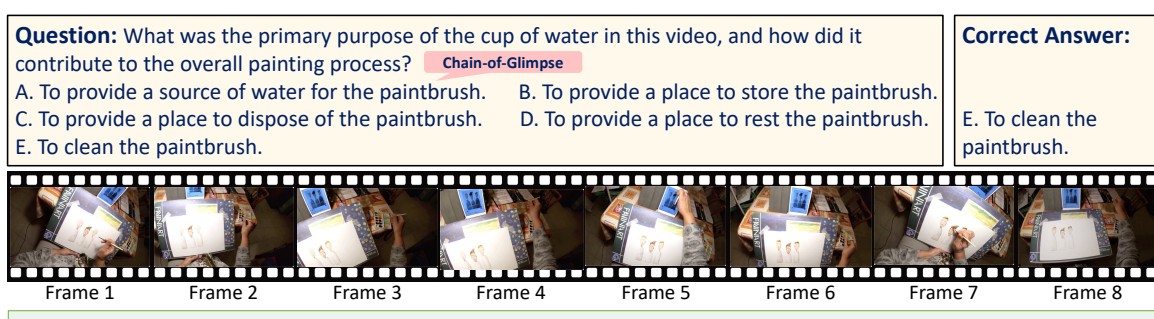

**Question:** What was the primary purpose of the cup of water in this video, and how did it contribute to the overall painting process? [Chain-of-Glimpse]
A. To provide a source of water for the paintbrush.    B. To provide a place to store the paintbrush.
C. To provide a place to dispose of the paintbrush.    D. To provide a place to rest the paintbrush.
E. To clean the paintbrush.

**Correct Answer:**

E. To clean the paintbrush.

Frame 1    Frame 2    Frame 3    Frame 4    Frame 5    Frame 6    Frame 7    Frame 8

**Chain-of-Glimpse**

<think>In Frame 2 and Frame 3, the cup of water is located at roughly (210, 560) on the table. The artist's hand is holding a paintbrush and moving between the palette at (450, 600) and the cup of water. Upon closer inspection of the artist's movements in Frame 4, the paintbrush is dipped into this cup to gather moisture before picking up pigment from the watercolor set at (450, 600). This interaction serves as the foundational step to activate the dry paints and adjust the consistency of the medium on the brush. Unlike simply resting or cleaning the tool, the continuous return to the water source is what allows the artist to maintain the fluid stroke necessary for the figures on the paper. Therefore, the cup of water serves as the essential source of moisture for the painting medium, corresponding to option A.</think><answer>A. To provide a source of water for the paintbrush.</answer>

*Figure 7.* Failure case.

from immediate interactions and does not capture subtle distinctions in action outcomes. Such cases highlight the challenge of reasoning over tool usage and intent beyond what is visually evident.

# E. Limitations and Future Work

While our Chain-of-Glimpse framework demonstrates effective object-grounded video reasoning, it currently relies on Qwen2.5-VL-3B and Qwen2.5-VL-7B as backbone models. These relatively small models have limited fundamental capabilities, which constrains the complexity of reasoning and the quality of visual-language alignment. In the future, we plan to explore larger multi-modal models with stronger representation power, incorporate more advanced pretraining, and investigate more efficient search strategies to scale the method to longer videos and more challenging tasks.

