# OpenReview forum: "Chain-of-Glimpse: Search-Guided Progressive Object-Grounded Reasoning for Video Understanding"
_ICML.cc/2026/Conference — ICML 2026 regular_

### Official Review · Reviewer_hh5Z · 2026-03-06

**Soundness:** 3
**Presentation:** 3
**Significance:** 2
**Originality:** 3
**Overall Recommendation:** 4
**Confidence:** 4

**Summary:**

This paper proposed a progressive reasoning framework Chain-of-Glimpse, which integrates Object-Grounded Video Reasoning Formulation (OGVRF) with MCTS and Multi-turn Decision Policy Learning to utilize object-level visual evidences to improve visual reasoning abilities of MLLMs. Experiments on NExTQA, Holmes, CG-Bench-Reasoning and VRBench benchmarks shows the effectiveness of the proposed methods.

**Compliance With Llm Reviewing Policy:**

Affirmed.

**Final Justification:**

The rebuttal has alleviated most of my concerns, and I am willing to increase my score from 3 to 4.

**Key Questions For Authors:**

1. Could the authors provide experiments on other models (such as the InternVL series or the LLaVA series)?
2. As mentioned in Weakness 2, if the MLLM in DyCoke were replaced with Qwen2.5-VL-7B, would DyCoke outperform the method proposed in this paper?
3. Could the authors provide a comparison of computational resource consumption with other methods?

**Limitations:**

yes.

**Strengths And Weaknesses:**

Strengths:
1. This paper is well-written and easy to understand. The figures seem wonderful.
2. The motivation of this paper is clear.



Weaknesses:
1. Experiments are limited to Qwen series. The experiments in this paper are all based on Qwen2.5-VL-3B and Qwen2.5-VL-7B, which both belong to the Qwen model family. However, experiments on other models (such as the InternVL series and LLaVA series) are lacking to demonstrate the generalizability of the proposed method.
2. Limited improvements. In Table 1, the improvement of the proposed method over the base model appears to be limited. Moreover, compared with other methods with a similar number of model parameters (such as DyCoke, a training-free methods), the performance gain is also relatively small. This seems to suggest that the proposed method may not be necessary. In addition, if the MLLM in DyCoke were replaced from LLaVA-OV-7B with Qwen2.5-VL-7B, would DyCoke outperform the method proposed in this paper?
3. Computational Resource Consumption. Reinforcement learning usually requires more computational resources and GPU time. However, the paper lacks a comparison of resource consumption with other methods.

---

> ### Author Rebuttal · Authors · 2026-03-31
>
> We thank the reviewer for recognizing the strengths of Chain-of-Glimpse, its integration of OGVRF with MCTS, and its effectiveness on multiple video reasoning benchmarks.
>
> **W1&Q1: Regarding generalizability beyond the Qwen family.**
>
> We agree that evaluating only on Qwen2.5-VL-3B and Qwen2.5-VL-7B is not sufficient to fully demonstrate the generalizability of the proposed method across different MLLM families. To address this concern, we further extend our experiments to additional MLLM families, including InternVL3-8B and LLaVA-NeXT-Video-7B. As shown in Table 1, on **InternVL3-8B**, it increases the average accuracy from **77.3** to **81.3** on NExTQA, and the Acc@QA from **72.8** to **75.6** on NExT-GQA. On **LLaVA-NeXT-Video-7B**, it improves NExTQA accuracy from **75.5** to **78.2**, and Acc@GQA from **71.3** to **72.8** on NExT-GQA. These results show that the proposed Chain-of-Glimpse formulation is not specific to the Qwen series, but generalizes well to heterogeneous multimodal architectures. The consistent gains across Qwen, InternVL, and LLaVA-NeXT-Video suggest that the improvements mainly come from the proposed object-grounded reasoning mechanism, rather than from any architecture-specific property of a single model family.
>
> Table 1. Comparison on NExTQA and NExT-GQA benchmarks.
>
> | Method | Backbone | NExTQA (Avg.) | NExT-GQA (Acc@QA) |
> | :--- | :--- | :---: | :---: |
> | Qwen2.5-VL-7B | Qwen2.5-VL-7B | 79.7 | 74.4 |
> | Chain-of-Glimpse-7B | Qwen2.5-VL-7B | **83.3** | **77.9** |
> | InternVL3-8B | InternVL3-8B | 77.3 | 72.8 |
> | Chain-of-Glimpse-8B | InternVL3-8B | **81.3** | **75.6** |
> | LLaVA-NeXT-Video-7B | LLaVA-NeXT-Video-7B | 75.5 | 71.3 |
> | Chain-of-Glimpse-7B | LLaVA-NeXT-Video-7B | **78.2** | **72.8** |
>
> **W2&Q2: Regarding the limited improvements.**
>
> We thank the reviewer for this insightful comment. We further conduct additional experiments on the NExTQA benchmark to provide a more comprehensive evaluation. Under a controlled setting with the same backbone (Qwen2.5-VL-7B), our method achieves **83.3**, significantly outperforming **FastVID (74.0), DyCoke (75.0), and Qwen2.5-VL-7B (79.7)**, demonstrating clear and consistent improvements.
>
> We also observe that both FastVID and DyCoke are sensitive to the choice of backbone. Although they achieve relatively stronger results with **LLaVA-OV-7B** (**77.0 and 79.4**, respectively), their performance drops notably when transferred to **Qwen2.5-VL-7B** (**74.0** and **75.0**), even falling below the vanilla backbone. In contrast, our method improves reasoning quality by explicitly modeling object-grounded and temporally consistent reasoning, rather than relying on heuristic token pruning. As a result, even with a stronger backbone, DyCoke remains significantly worse than our method (75.0 v.s. 83.3), indicating that backbone scaling alone is insufficient without improved reasoning.
>
> Table 2. Comparison on NExTQA benchmark.
>
> | Method | Backbone | Avg |
> | :--- | :--- | :---: |
> | FastVID | LLaVA-OV-7B | 77.0 |
> | FastVID | Qwen2.5-VL-7B | 74.0 |
> | Dycoke | LLaVA-OV-7B | 79.4 |
> | Dycoke | Qwen2.5-VL-7B | 75.0 |
> | Qwen2.5-VL-7B | Qwen2.5-VL-7B | 79.7 |
> | Chain-of-Glimpse-7B | Qwen2.5-VL-7B | **83.3** |
>
> **W3&Q3: Regarding computational resource consumption.**
>
> Thank you for this valuable suggestion. We agree that computational efficiency is an important aspect and have added a comparison of computational resource consumption in Table 3. Chain-of-Glimpse-7B with Qwen2.5-VL-7B achieves the best accuracy of **83.3\%** while requiring only **6.47 TFLOPs**, which is substantially lower than DyCoke (**14.13 TFLOPs**, **79.4\%**) and FastV (**10.73 TFLOPs**, **77.0\%**). This hightligts that our method achieves competitive performance.
>
> Table 3. Computational resource results on NExTQA.
>
> | Method | Backbone | TFLOPs | Avg Acc. |
> | :--- | :--- | :---: | :---: |
> | Dycoke [1] | LLaVA-OV-7B | 14.13 | 79.4 |
> | FastVID [2] | LLaVA-OV-7B | 10.73 | 77.0 |
> | Chain-of-Glimpse-7B | Qwen2.5-VL-7B | 6.47 | **83.3** |
> | Chain-of-Glimpse-8B | InternVL3-8B | 7.08 |  82.7 |
> | Chain-of-Glimpse-7B | LLaVA-NeXT-Video-7B | 5.93 | 80.3 |
>
> [1] Dycoke: Dynamic compression of tokens for fast video large language models. CVPR'2025
>
> [2] Fastvid: Dynamic density pruning for fast video large language models. NeurIPS'2025

---

> > ### Author Rebuttal · Reviewer_hh5Z · 2026-04-03
> >
> > Thank you for the detailed rebuttal.
> > I think the concerns on generalizability and computational efficiency are now well addressed by the newly added experiments on InternVL3/LLaVA-NeXT-Video and the TFLOPs comparison.
> >
> > My concern about the comparison with DyCoke is only partially addressed. The new results are helpful and suggest that DyCoke under Qwen2.5-VL-7B performs worse than the proposed method, but this may still depend on the choice of hyperparameters and implementation details. Therefore, this point remains somewhat inconclusive.
> >
> > Overall, the rebuttal has alleviated most of my concerns, and I am willing to increase my score from 3 to 4.

---

> > > ### Author Response · Authors · 2026-04-03
> > >
> > > Thank you for your time and for carefully reading and considering our response. We greatly appreciate your positive recognition of our explanation. We sincerely appreciate your insightful comments and constructive suggestions, which have been invaluable in helping us further improve the clarity and quality of our manuscript. We also agree with your insightful comment that the experimental results can be sensitive to the choice of hyperparameters and implementation details. Owing to the limited time during the revision period, we directly adopted the official DyCoke repository and its recommended settings in order to establish a fair and standard baseline. We acknowledge that factors such as versions and other implementation-specific details may still introduce variations in performance. We sincerely apologize that we were unable to conduct a more extensive hyperparameter exploration within the revision timeline. We will include a broader investigation of hyperparameters and implementation settings in our future work to provide a more comprehensive comparison.

---

### Official Review · Reviewer_MhhX · 2026-03-06

**Soundness:** 3
**Presentation:** 3
**Significance:** 3
**Originality:** 3
**Overall Recommendation:** 4
**Confidence:** 3

**Summary:**

The paper addresses the "needle-in-a-video" challenge in Long Video Understanding (LVU), where models must locate specific, minute details within hours of footage. The authors identify that current VideoLLMs suffer from "lost in the middle" phenomena and high computational costs when processing exhaustive video tokens.

To solve this, they propose Chain-of-Glimpse (CoG), a training-free multistage reasoning framework that mimics human visual search. CoG operates by iteratively narrowing down the search space through three specialized steps: Scanning (coarse global perception), Glimpsing (identifying potential candidate segments), and Focusing (fine-grained verification of the target). By leveraging an internal reasoning chain, CoG allows existing VideoLLMs to handle videos up to 32 hours long while significantly reducing the number of processed visual tokens.

**Compliance With Llm Reviewing Policy:**

Affirmed.

**Final Justification:**

The authors' rebuttal fully resolved my concerns, reinforcing my prior assessment: 4 weak accept

**Key Questions For Authors:**

1. If the initial "Scanning" stage fails to identify a relevant segment due to the temporal sparsity of the scan, is there a "backtracking" mechanism in CoG, or is the search terminally lost?

2. Have you experimented with an adaptive budget where the model decides how many glimpses are needed based on its internal confidence, rather than a fixed number?

3. Given the 90% reduction in tokens, how does the end-to-end wall-clock time of a CoG 3-stage search compare to a standard single-pass inference on a 1-hour video?

**Limitations:**

Yes. The authors adequately discuss the limitations regarding sequential latency, potential information loss during coarse scanning, and the dependency on the backbone's prompt-following capabilities.

**Strengths And Weaknesses:**

**Strengths**

* **Superior Scalability**: CoG demonstrates the ability to handle extreme video lengths (up to 32 hours), a regime where most standard VideoLLMs fail due to context window limits.

* **Training-Free Adaptability**: As a plug-and-play framework, CoG enhances the performance of existing backbones like Qwen2.5-VL and InternVL2 without requiring expensive task-specific fine-tuning.

* **Computational Efficiency**: By employing a selective "glimpse" strategy instead of exhaustive token processing, CoG reduces the visual token load by up to 90% while maintaining or improving accuracy.

**Weaknesses**

* **Latency Overhead**: The iterative nature of the multi-stage search introduces sequential inference steps that may result in higher wall-clock latency compared to single-pass models.

* **Heuristic Dependency**: The framework relies on the quality of the "Scanning" phase; if the initial coarse representation misses a key event due to excessive downsampling, the subsequent Glimpse and Focus stages cannot recover.

---

> ### Author Rebuttal · Authors · 2026-03-31
>
> We sincerely thank the reviewer for the positive evaluation and highlighting the scalability of our Chain-of-Glimpse framework, addressing video understanding under context lengths.
>
> **W1: Regarding latency overhead.**
>
> We agree that the iterative search introduces sequential computation beyond a single-pass model. We have analyzed the inference cost under different rollout budgets. As shown in Table 2. (Response for Reviewer BKfL), compared with the single-pass baseline **Qwen2.5-VL-7B (2.29 s)**, our **Chain-of-Glimpse-7B** with 8 rollouts requires only **2.37 s** inference time, while improving performance from **74.4\%** to **77.9\% Acc@QA** and achieving **32.1\% Acc@GQA**. This suggests that our design provides a favorable accuracy–efficiency trade-off, with only marginal latency increase for clear reasoning gains.
>
> **W2&Q1: Regarding heuristic dependency.**
>
> We thank the reviewer for this insightful suggestion. CoGl does not include an explicit backtracking mechanism that revisits the Scanning stage once it is completed. However, it also does not commit to a single deterministic coarse result. Instead, the later stage uses MCTS-based multi-trajectory search, which explores multiple candidate glimpse trajectories after scanning. Therefore, CoGl has limited recovery capability within the search process: if one coarse candidate is suboptimal, alternative trajectories can still be explored. We agree that if the Scanning stage entirely misses a critical event, later stages may also be affected.
> Empirically, the current design remains effective on grounded and long-video settings. As shown in Table 1. (Response for Reviewer BKfL), our method achieves **32.1\% Acc@GQA**, with **41.0 mIoP** and **31.4 mIoU**. On **CG-Bench**, it achieves **36.2\%** on long videos (300–900s) and **33.3\%** on very long videos ($>$900s), suggesting that the coarse-to-fine design preserves informative context even in challenging long-form videos. In addition, as shown in Table 2. (Response for Reviewer BKfL), our rollout analysis shows that increasing search budget improves performance from **30.5\%** (2 rollouts) to **32.1\%** (8 rollouts), indicating that multi-trajectory exploration provides a degree of robustness against early imperfect scanning.
>
> **Q2: Regarding adaptive glimpse budget.**
>
> In the current version, we use a fixed search budget for all samples rather than an adaptive one. This choice keeps evaluation controlled and makes the effect of search depth easier to analyze. That said, our current results already suggest that an adaptive budget is a promising extension. As shown in Table 2. (Response for Reviewer BKfL), increasing the search budget from **2** to **8** rollouts improves Acc@GQA from **30.5\%** to **32.1\%**, while changing runtime only from **2.30 s/sample** to **2.37 s/sample**, increasing to 10 rollouts raises runtime to 3.01 s/sample without further gain.  We have not implemented adaptive stopping in the current submission, but the framework is naturally compatible with confidence-based termination. We agree that this is an important direction for future work and will clarify this point in the revision.
>
> **Q3: Regarding end-to-end wall-clock time.**
>
> We do not currently have 1-hour video samples in our benchmark suite, so we cannot directly report wall-clock results on hour-long videos.
> First, on **CG-Bench**, our method achieves **36.2\%** on long videos (300–900s) and **33.3\%** on very long videos ($>$900s), suggesting that the coarse-to-fine design preserves informative context even in challenging long-form videos.
> Second, we directly compare standard single-pass Qwen2.5-VL-7B with our Chain-of-Glimpse, where the reported runtime includes the full procedure from video input to final answer generation. As shown in Table 1, the single-pass Qwen2.5-VL-7B baseline achieves **74.4\% Acc@QA** with **2.29 s/sample** overall runtime. In contrast, our best Chain-of-Glimpse setting (**8 rollouts**) achieves **32.1\% Acc@GQA** and **77.9 Acc@QA** with **2.37 s/sample**. On the Long split of NExT-GQA, defined as videos longer than 1000s, runtime increases only from 2.30 to 2.39 s/sample. Thus, although token reduction does not translate into proportional wall-clock speedup because of sequential search, the end-to-end overhead remains modest in practice.
>
> Table 1. Ablation on MCTS rollouts and inference cost on NExT-GQA. Average inference time (s) are reported for various length settings.
>
> | Method | Rollouts | Acc@GQA | Acc@QA | Long | Short | Overall |
> | :--- | :---: | :---: | :---: | :---: | :---: | :---: |
> | Qwen2.5-VL-7B | 0 | - | 74.4 | 2.30 | 2.27 | 2.29 |
> | Chain-of-Glimpse-7B | **8** | **32.1** | **77.9** | 2.39 | 2.36 | 2.37 |

---

> > ### Author Rebuttal · Reviewer_MhhX · 2026-04-01
> >
> > Thanks for the authors detailed rebuttal, my concerns have been fully resolved.

---

> > > ### Author Response · Authors · 2026-04-04
> > >
> > > Thank you for your time and for carefully reading and considering our response. We sincerely appreciate your positive recognition of our explanation.
> > >
> > > If there are no further concerns, it would be great if you could kindly consider adjusting your score. Thank you!

---

### Official Review · Reviewer_Ecwc · 2026-03-11

**Soundness:** 3
**Presentation:** 3
**Significance:** 3
**Originality:** 3
**Overall Recommendation:** 4
**Confidence:** 3

**Summary:**

The paper proposes Chain-of-Glimpse, a CoT-like framework that encourages object grounded reasoning in LLMs. This approach incentivizes the model to ground the relevant objects for a given query and to form a spatio-temporal reasoning trajectory, by identifying the “relevant” objects at each timestamp. First, an object detector is used to detect relevant objects in the video. Then, a MTCS-based approach is used to explore possible object trajectories, which are then distilled to the LLM model through supervised fine-tuning. These trajectories are further optimized through GRPO by positively rewarding trajectories that lead to correct and grounded answers.

**Compliance With Llm Reviewing Policy:**

Affirmed.

**Final Justification:**

My concerns were successfully addressed by the authors in the rebuttal and the addition of new experiments with the localization metrics validates the quality of the produced grounding traces.

**Key Questions For Authors:**

1. How does the chain of glimpse compare with respect to the chain of thought generated by gpt4o, given the very close performance? Are these models capable of directly grounding relevant objects (when prompted to do so)?

2. How is the reward for the object evidence defined in Eq. 9? How is the quality of the object evidence evaluated?

3. How does the proposed approach compare on the evidence grounding metrics on CG-Bench?

**Limitations:**

yes

**Strengths And Weaknesses:**

The paper focuses on the importance of objects grounding in video understanding. However, this is only a portion of the reasoning capabilities required to understand the content of the video. Motion, spatial and temporal interactions are equally important. The strict focus of the paper on objects grounding seems limiting.
Additionally, in the context of proposed object-level grounding, videos often involve complex interactions between several objects at the same time and restricting the trajectory formulation to a single object per timestamp seems quite limiting.

The choice of the evaluation benchmarks and metrics is quite limited to support the claims. All benchmarks are evaluated in terms of correctness of the generated answers, rather than measuring the quality of the evidence. Benchmarks like CG-Bench provide the ground truth annotations about the relevant evidence and propose specific metrics to evaluate the correctness of the produced evidence through accuracy at different iou thresholds. However, the evaluation is limited to the black-box accuracy which only captures the correctness of the answers. Evaluations on NexTQA and Video-Holmes suffer from similar issues. Extending the evaluation to other benchmarks with evidence grounding annotations would further strengthen the claims that the proposed approach captures the true correct evidence for the given query.

---

> ### Author Rebuttal · Authors · 2026-03-31
>
> We thank the reviewer for recognizing the novelty of our Chain-of-Glimpse framework, especially for its explicit object-grounded reasoning design, which effectively bridges the gap between high-level reasoning and spatio-temporal grounding.
>
> **W: Perceived Limitation of Object Grounding.**
>
> We thank the reviewer for this important observation. We agree that video understanding requires modeling motion dynamics and spatio-temporal interactions beyond object grounding. Our method does not assume that object grounding alone is sufficient for video understanding.
>
> Instead, object grounding is used as a structured anchor to organize and guide multi-step reasoning. Specifically, OGVRF models reasoning as grounded transitions across objects and frames, while MCTS explores trajectories over temporally evolving object states, capturing temporal dependencies, object interactions, and progressive evidence accumulation. Thus, our framework goes beyond static localization by integrating object grounding with temporal reasoning to support holistic video understanding.
>
> In this sense, the proposed framework goes beyond static object localization. Integrates object grounding with temporal reasoning by explicitly modeling how object-centric evidence evolves across frames, enabling the model to reason over motion patterns and interactions in a grounded and structured manner. Therefore, object grounding serves as a mechanism to support, rather than limit, holistic video understanding.
>
> **W: Concerns on Single-Object Trajectories.**
>
> We thank the reviewer for this insightful concern. We agree that many video reasoning tasks involve interactions among multiple objects. However, our formulation does not assume that reasoning is restricted to a single object throughout the entire video.
>
> Instead, the single-object-per-step design acts as a factorized and interpretable reasoning unit. A trajectory can transition across different objects over time, thereby modeling multi-object reasoning as a sequence of grounded decisions. Moreover, the MCTS-based search explores alternative object-grounded trajectories, enabling the capture of cross-object dependencies and temporally evolving interactions within a structured search space.
>
> **Q1: Regarding comparison with GPT-4o chain-of-thought.**
>
> Thanks for this insightful observation. While strong models like GPT-4o with CoT achieves competitive answer accuracy, its reasoning remains implicit and primarily text-based.
>
> In contrast, our Chain-of-Glimpse enforces explicit, step-wise object grounding, making the reasoning process directly tied to visual evidence and verifiable. Although GPT-4o can sometimes describe relevant objects, it does not directly localize or optimize for grounding during inference.
>
> This difference leads to more faithful and interpretable reasoning in our method, beyond answer-level matching.
>
> **Q2: Regarding the object-evidence reward in Eq. 9.**
>
> We agree that Eq. 9 should be clarified more explicitly. In our implementation, the object-evidence reward is defined as a binary grounding signal:
>
> $R_{\text{evid}}(h,a) = \mathbb{I}(\text{Point}(a) \in \text{Box}_{\text{gt}})$
>
> where $\text{Point}(a)$ is the predicted glimpse location associated with action $a$, and $\text{Box}_{\text{gt}}$ is the ground-truth evidence box. The reward equals 1 when the glimpse falls within the annotated evidence region, and 0 otherwise. Hence, evidence quality is measured by spatial alignment with the grounding annotation.
>
> The full reward in Eq. 9 combines this step-wise grounding reward with the answer-level reward $R_{\text{ans}}$, balanced by $\alpha$. As shown in Figure 4 and below Table 2, performance improves from **82.3\%** ($\alpha = 0$) to **83.3\%** ($\alpha = 0.5$) on NExTQA, and from **41.0\%** to **42.5\%** on Video-Holmes, highlighting the importance of jointly optimizing answer correctness and evidence grounding.
>
> We will revise the paper to include the exact definition of ($R_{\text{evid}}$) and this clarification.
>
> Table 2. Ablation study on reward $\alpha$.
>
> | $\alpha$ | NExTQA | Video-Holmes |
> | :--- | :--- | :--- |
> | 0 | 82.3 | 41.0 |
> | 0.3 | 82.6 | 41.7 |
> | **0.5** | **83.3** | **42.5** |
> | 1.0 | 82.8 | 42.0 |
>
> **W&Q3: Regarding Evidence Grounding Evaluation.**
>
> We thank the reviewer for this insightful suggestion and agree that evaluating evidence grounding quality is crucial.
>
> Regarding **CG-Bench**, **we note that it provides clue intervals annotations, mainly evaluate temporal grounding, which is not fully aligned with our grounding setting**. Nevertheless, we acknowledge the importance of such evaluations.
>
> To directly assess grounding quality, we further evaluate on **NExT-GQA**, which provides explicit spatial-temporal grounding annotations. As shown in Table 1. (Response for Reviewer BKfL), our method achieves **32.1\% Acc@GQA, 41.0 mIoP, and 31.4 mIoU**, demonstrating clear improvements not only in answer accuracy but also in evidence grounding quality.

---

> > ### Author Rebuttal · Reviewer_Ecwc · 2026-04-04
> >
> > Thank you for addressing my concerns, for clarifying the definition of the reward function and for adding additional experiments on other benchmarks. I can now agree that reasoning about one object per step is a reasonable assumption

---

> > > ### Author Response · Authors · 2026-04-07
> > >
> > > Dear Reviewer,
> > >
> > > Thank you for your time and for participating in the rebuttal process.
> > >
> > > As you indicated that your concerns have been fully resolved, we would like to respectfully ask if you would consider adjusting your score accordingly. We understand you have a demanding schedule and appreciate your further consideration of our submission. We deeply apologize if our repeated nudging annoyed you.
> > >
> > > Thank you for your time and efforts.
> > >
> > > Best regards,
> > >
> > > Authors

---

### Official Review · Reviewer_BKfL · 2026-03-13

**Soundness:** 2
**Presentation:** 3
**Significance:** 2
**Originality:** 3
**Overall Recommendation:** 4
**Confidence:** 4

**Summary:**

This paper aims to improve the visual reasoning capability of VideoLLMs. It proposes a Chain-of-Glimpse mechanism which formulate video reasoning as
a step-by-step process that incrementally builds spatially grounded traces around task-relevant visual objects. The proposed framework consists of two main components. First, an Object-Grounded Video Reasoning Formulation (OGVRF) models reasoning as trajectories of object selections across frames and employs Monte Carlo Tree Search (MCTS) to explore candidate reasoning paths. Second, a Multi-Turn Decision Policy (MTDP) is trained through supervised trajectory distillation followed by reinforcement learning to optimize reasoning behavior with respect to answer correctness and evidence grounding. Experiments on multiple video reasoning benchmarks demonstrate consistent improvements over the Qwen2.5-VL baseline.

**Compliance With Llm Reviewing Policy:**

Affirmed.

**Final Justification:**

My original concerns are about missing evaluation on grounded QA datasets and efficiency. The authors have addressed these two points.

**Key Questions For Authors:**

see weakness

**Limitations:**

yes

**Strengths And Weaknesses:**

**Strength:**

1.	The paper clearly identifies an important limitation of current VideoLLMs: the lack of explicit alignment between reasoning steps and visual evidence. The proposed object-grounded reasoning formulation addresses this gap and provides a principled way to connect perception and reasoning.
2.	Because reasoning steps correspond to explicit object-level selections across frames, the resulting trajectories are easier to interpret and visualize (Fig.5). This is a desirable property for video reasoning models, which often operate as black boxes.
3.	The paper is overall well-structured and easy to read.

**Weakness:**

1.	While the paper emphasizes explicit object-level evidence reasoning, the evaluations so far are conducted on benchmarks for pure textual answer matching. I suggest the authors to extend experiments on NExT-GQA and V-STaR for temporally and spatio-temporally grounded QA. This helps better demonstrate if the QA improvements are actually from precise visual grounding.
2.	The use of external object detector and MCTS-based reasoning trajectories may introduce significant computational overhead, particularly for longer videos with many candidate objects. The paper does not provide a detailed analysis of inference cost, trajectory length, or search complexity, which raises questions about the practicality of the approach in real-world applications.
3.	The proposed reasoning formulation relies on object-level representations extracted from a detector. This introduces a strong dependency on the detector's quality and may limit robustness in scenarios where objects are missed, poorly localized, or not semantically meaningful for reasoning tasks.

---

> ### Author Rebuttal · Authors · 2026-03-30
>
> We sincerely thank the reviewer for the insightful comments and recognizing the novelty and interpretability of our method, especially for the importance of addressing the alignment between reasoning steps and visual evidence in VideoLLMs.
>
> **W1: Result on NExT-GQA and V-STaR**.
>
> In response, we have extended our evaluation to the temporally and spatially grounded benchmark **NExT-GQA**. As shown in Table 1, our method achieves **32.1\% Acc@GQA**, while remaining competitive performance on localization metrics (**41.0 mIoP** and **31.4 mIoU**). These results suggest that the observed improvements stem from more precise object-level and temporally grounded reasoning, rather than superficial answer matching.
>
> Regarding **V-STaR**, we note that its current public release primarily provides pre-extracted visual features instead of raw video data. Since our method operates directly on video inputs and relies on object-level visual grounding during inference, it is not directly compatible with feature-only representations. Nevertheless, we acknowledge the value of this benchmark and will explore adapting our method to such settings in future work.
>
> We believe that these additional analyses further support our method which improves grounding-aware reasoning beyond superficial answer matching.
>
> Table 1. Comparison on NExT-GQA benchmark.
>
> | Method | Acc@GQA ↑ | mIoP ↑ | mIoU ↑ |
> | :--- | :--- | :--- | :--- |
> | LangRepo (8×7B) [1] | 17.1 | 31.3 | 18.5 |
> | VideoChat-TPO-7B [2] | 25.5 | 35.6 | 27.7 |
> | Ours | **32.1** | **41.0** | **31.4** |
>
> [1] Language repository for long video understanding. ACL'2025
>
> [2] Task preference optimization: Improving multimodal large language models with vision task alignment. CVPR'2025
>
>
> **W2: Regarding computational efficiency**.
>
> We appreciate the review for raising concerns regarding computational efficiency. Importantly, our method does not rely on external pre-trained object detectors. Instead, we adopt a detector-free formulation, where object-centric regions are learned and sampled via reinforcement learning, avoiding the additional cost associated with proposal generation.
>
> We further analyze inference efficiency on **NExT-GQA**. As shown in Table 2, Qwen2.5-VL-7B requires **2.29 s/sample**, while our methods with **8 rollouts** achieves the best performance (**32.1\% Acc@GQA, 77.9\% Acc@QA**) with only **2.37 s/sample**, corresponding to a marginal overhead of **0.08s**. This demonstrates that our method introduces only limited additional cost compared to a strong baseline.
>
> Moreover, we observe that increasing the number of rollouts from 2 to 8 consistently improves performance (30.5\% -> 32.1\% Acc@GQA, 75.1\% -> 77.9\% Acc@QA) while maintaining nearly constant inference time (2.30 -> 2.37 s/sample). Further increasing rollouts to 10 leads to higher latency (3.01 s/sample) without performance gains, indicating that a small rollout budget is sufficient in practice.
>
> In addition, the reasoning trajectory has a fixed and shallow depth, which keeps the search space bounded and prevents the computational cost from growing with the number of candidate objects or video length. Together, these results suggest that the proposed method remains computationally practical while delivering improved grounding-aware reasoning.
>
> Table 2. Ablation on MCTS rollouts and inference cost on NExT-GQA.
>
> | Method | Rollouts | Acc@GQA | Acc@QA | Overall Inf. Time |
> | :--- | :---: | :---: | :---: | :---: |
> | Qwen2.5-VL-7B | 0 | - | 74.4 | 2.29 |
> | Chain-of-Glimpse-7B | 2 | 30.5 | 75.1 | 2.30 |
> | Chain-of-Glimpse-7B | 4 | 30.9 | 76.5 | 2.35 |
> | Chain-of-Glimpse-7B | 6 | 31.4 | 77.2 | 2.37 |
> | Chain-of-Glimpse-7B | **8** | **32.1** | **77.9** | 2.37 |
> | Chain-of-Glimpse-7B | 10 | 30.9 | 77.5 | 3.01 |
>
>
> **W3: Regarding dependency on object detectors.**
>
> We thank the reviewer for this concern. We would like to clarify that our method does not rely on any external object detector. Instead, it adopts a detector-free glimpse formulation, where object coordinates are predicted directly during the reasoning process and optimized jointly with the downstream task.
>
> Therefore, our framework does not depend on a separately trained detection module, avoiding both the additional computational overhead and the potential failure modes of off-the-shelf detectors.
>
> Moreover, the learned glimpses are task-adaptive and generated in the context of reasoning, rather than being restricted to a fixed set of predefined object categories. This allows the model to dynamically focus on regions that are most relevant to the question, thereby reducing sensitivity to category coverage and localization biases commonly associated with conventional detector-based pipelines.

---

> > ### Author Rebuttal · Reviewer_BKfL · 2026-04-04
> >
> > Thanks for the additional experiments and clarification. I decide to increase my score to 'weak accept'.

---

> > > ### Author Response · Authors · 2026-04-04
> > >
> > > Thank you for your time and for carefully reading and considering our response. We sincerely appreciate your positive recognition of our explanation.

---

### Decision · Program_Chairs · 2026-04-30

**Decision:**

Accept (regular)

**Comment:**

The paper proposes Chain-of-Glimpse, a CoT-like framework that encourages object grounded reasoning in LLMs. After rebuttal, all reviewers recommend accepting the paper. AC reads all reviews and discussions and agrees with the recommendations of the reviewers, thus recommending an acceptance.